# Open-Reasoner-Zero: An Open Source Approach to Scaling Up Reinforcement Learning on the Base Model

**Jingcheng Hu**[12*]  **Yinmin Zhang**[2]   **Qi Han**[2]
**Daxin Jiang**[2]   **Xiangyu Zhang**[2]    **Heung-Yeung Shum**[1]
[1]Tsinghua University    [2]StepFun
{hujingcheng, zhangyinmin, hanqer}@stepfun.com
GitHub: https://github.com/Open-Reasoner-Zero/Open-Reasoner-Zero
HuggingFace: https://huggingface.co/Open-Reasoner-Zero.

## Abstract

We introduce Open-Reasoner-Zero, the first open source implementation of large-scale reasoning-oriented RL training on the base model focusing on scalability, simplicity and accessibility. Through extensive experiments, we demonstrate that a minimalist approach, vanilla PPO with GAE ($\lambda = 1$, $\gamma = 1$) and straightforward rule-based rewards, without any KL regularization, is sufficient to scale up both benchmark performance and response length, replicating the scaling phenomenon observed in DeepSeek-R1-Zero. Using the same base model, Qwen2.5-32B base, as DeepSeek-R1-Zero-Qwen-32B, our implementation achieves superior performance across AIME2024, MATH500, and GPQA Diamond, while demonstrating remarkable efficiency—requiring only 1/10 of the training steps compared to the DeepSeek-R1-Zero pipeline. We validate that this recipe generalizes well across diverse training domains and different model families without algorithmic modifications. Moreover, our analysis not only covers training dynamics and ablation for critical design choices, but also quantitatively shows how the learned critic in Reasoner-Zero training effectively identifies and devalues repetitive response patterns, yielding more robust advantage estimations and enhancing training stability. Embracing the principles of open-source, we release our source code, training data, and various model weights, fostering reproducibility and encouraging further exploration of the properties of related models.

## 1   Introduction

Large-scale reinforcement learning (RL) training of language models on reasoning tasks has emerged as a promising paradigm for mastering complex problem-solving skills. Recent breakthroughs, particularly OpenAI's o1 [1] and DeepSeek's R1-Zero [2], have demonstrated remarkable training time scaling: as the training computation scales up, both the model's benchmark performance and response length consistently and steadily increase without any sign of saturation. Inspired by these advancements, we aim to explore this new scaling phenomenon by conducting large-scale RL training, even applying it directly to base models, an approach we refer to as Reasoner-Zero training.

In this work, we introduce Open-Reasoner-Zero (ORZ), the first open-source implementation of large-scale reasoning-oriented RL training on large language models (LLMs) with our empirical best practices, designed to be robust, scalable and simple-to-follow. Under Reasoner-Zero paradigm,

---

*   The work was done during Jingcheng Hu's internship at StepFun. This research is supported in part by the National Key Research and Development Program of China (Grant No. 2023ZD0121300), and National Natural Science Foundation of China (Grant No. 62495092)

39th Conference on Neural Information Processing Systems (NeurIPS 2025).

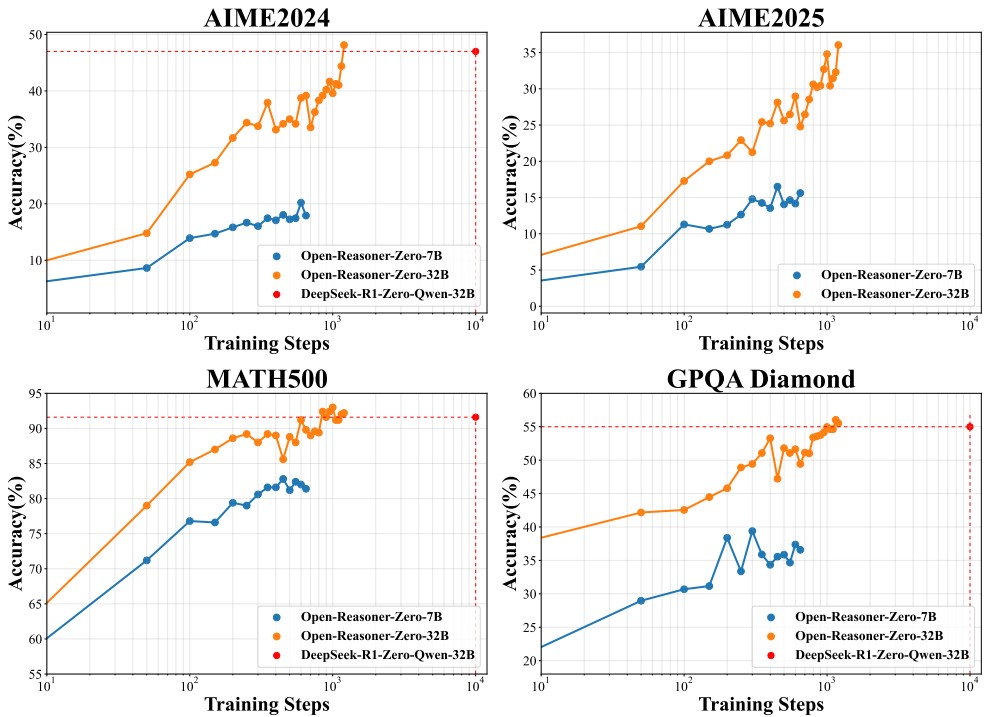

Figure 1: Evaluation performance of Open-Reasoner-Zero-{7B, 32B} on benchmarks (averaged on 16 responses) during training. Using the same base model, Qwen2.5-32B base, as DeepSeek-R1-Zero-Qwen-32B, Open-Reasoner-Zero-32B achieves superior performance on AIME2024, MATH500, and GPQA Diamond benchmarks—requiring only a tenth of the training steps.

LLMs are trained to master diverse reasoning skills under verifiable rewards, spanning arithmetic, logic, coding and common-sense reasoning (*e.g.*, scientific problems, numerical reasoning, natural language understanding and even creative writing). While DeepSeek's R1-Zero outlined their training pipeline briefly, we provide a comprehensive study of our training strategy, with in-depth insights into overcoming training instability from value estimation perspectives in RL. Our goal is to democratize advanced RL training techniques accessible to the broader research community.

Our proposed Open-Reasoner-Zero, built on the same Qwen2.5-32B base model as DeepSeek-R1-Zero-Qwen-32B, achieves superior performance on challenging benchmarks including AIME24, MATH500, and GPQA Diamond, while requiring only 1/10 of the training steps. Through extensive ablation studies, we summarize some key findings. Specifically, we find that vanilla PPO using GAE ($\lambda = 1$ and $\gamma = 1$) and without any KL-related regularization, combined with a straightforward rule-based reward, is sufficient to achieve steady scalability in both benchmark performances and response length across varying model sizes, different model families, and diverse task distributions, when trained on large-scale carefully curated datasets. Furthermore, we investigate several critical aspects of Reasoner-Zero training: (1) training dynamics, including how performance and response length evolve throughout training; (2) value and advantage estimation effectiveness, illustrating how PPO's learned critic model leads to robust advantage estimates; and (3) comprehensive ablation studies on key design choices. These investigations provide valuable insights into the mechanisms behind successful large-scale reasoning-oriented RL training.

Our primary contributions are as follows:

1. We provide a fully open-source implementation of large-scale RL training directly on a base LLM, a strategy we refer to as Open-Reasoner-Zero.

2. We present novel insights crucial for achieving stable and scalable Reasoner-Zero training, encompassing key findings regarding effective design choices, alongside a thorough investigation for advantage estimation.

3. We validate the generalizability of our recipe across different model sizes, various model families and diverse task domains.

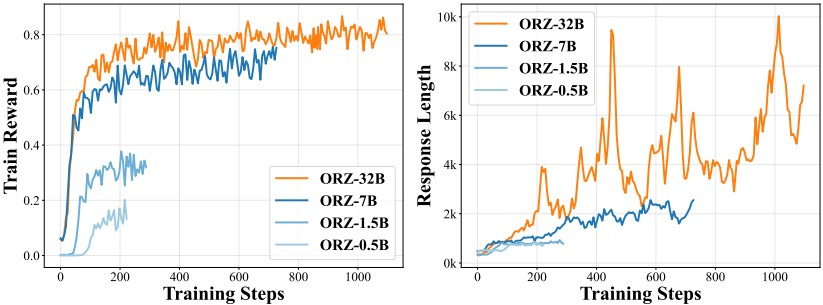

Figure 2: Train-time Scale up on Train Reward and Response Length of Open-Reasoner-Zero (ORZ) - {0.5B, 1.5B, 7B, 32B}. Train Reward and Response Length increase steadily, demonstrating consistent scalability across model sizes. Interestingly, the ORZ-32B Response Length exhibits fluctuations without negatively impacting training stability, highlighting the robustness of our minimalist recipe.

4. We release comprehensive resources including code, data, and model to the community.

## 2 Scale-up Reinforcement Learning from a Base Model

In this section, we describe the strategy and critical components for scale-up reasoning-oriented RL directly from a base model. Concretely, we begin by reviewing essential background on Generalized Advantage Estimation (GAE) [3] and Proximal Policy Optimization (PPO) [4] algorithms. Subsequently, we discuss key insights derived from our comprehensive ablation experiments that enable successful scale-up RL training. Finally, we detail the fundamental yet critical implementation settings for our approach, covering data curation, prompt design, and reward function specification.

### 2.1 RL Algorithm

We adopt the PPO [4] as the core RL algorithm, diverging from the GRPO used in DeepSeek-R1-Zero [2]. Specifically, for each input question $q$ (*i.e.,* prompt), the policy model generates a group of responses $\{o_1, o_2, ..., o_n\}$, where $n$ represents the number of sampled responses (*i.e.*, rollout size per prompt). Each response $o_i$ constitutes a trajectory $\tau_i = (s_0, a_0, ..., s_{T_i-1}, a_{T_i-1})$, where $s_t$ is the state (prompt + previously generated tokens) and $a_t$ is the token generated at step $t$ (*i.e.*, token $t$). Using our rule-based reward function, each trajectory receives a single terminal reward $R_i \in \{0, 1\}$, assigned at the end of the sequence ($r_t = 0$ for $t < T_i - 1$, $r_{T_i-1} = R_i$).

We utilize GAE [3] to estimate the advantage $\hat{A}_t$ for each token. The general GAE formula is:

$$\hat{A}_t^{GAE(\gamma,\lambda)} = \sum_{k=0}^{T-t-1} (\gamma\lambda)^k \delta_{t+k}, \tag{1}$$

where $\delta_{t+k} = r_{t+k} + \gamma V_\phi(s_{t+k+1}) - V_\phi(s_{t+k})$ is the Temporal Difference (TD) error, $V_\phi$ is the value function parameterized by $\phi$, $\gamma$ is the discount factor, and $\lambda$ controls the bias-variance trade-off.

The general PPO objective updates the policy parameters $\theta$ to maximize a clipped surrogate objective function, and the value parameters $\phi$ to minimize the error between the value estimate $V_\phi(s_t)$ and a target value $V_t^{\text{target}}$, typically the discounted return. The standard objectives are:

$$\mathcal{J}_{\text{PPO}}(\theta) = \mathbb{E}_{\tau \sim \pi_{\theta_{\text{old}}}} \left[ \sum_{t=0}^{T-1} \min\left( \rho_t(\theta)\hat{A}_t, \text{clip}(\rho_t(\theta), 1-\epsilon, 1+\epsilon)\hat{A}_t \right) \right], \tag{2}$$

$$\mathcal{J}_{\text{value}}(\phi) = \frac{1}{2}\mathbb{E}_{\tau \sim \pi_{\theta_{\text{old}}}} \left[ \sum_{t=0}^{T-1} (V_\phi(s_t) - V_t^{\text{target}})^2 \right], \tag{3}$$

where $\rho_t(\theta) = \frac{\pi_\theta(a_t|s_t)}{\pi_{\theta_{\text{old}}}(a_t|s_t)}$ is the probability ratio, and the clipping parameter $\epsilon$ is set to 0.2 in our cases. Commonly, $V_t^{\text{target}}$ is the estimated discounted return $G_t = \hat{A}_t^{GAE(\gamma,\lambda)} + V_\phi(s_t)$.

Table 1: Comparison of Open-Reasoner-Zero-32B with DeepSeek-R1-Zero-Qwen-32B DAPO-Qwen-32B on reasoning-related benchmarks. DeepSeek-R1-Zero-Qwen-32B results are from [2]. DAPO-Qwen-32B[*] results were obtained using our evaluation metric on the released checkpoint.

| Model | AIME 2024 | AIME 2025 | MATH500 | GPQA Dia. |
|---|---|---|---|---|
| DeepSeek-R1-Zero-Qwen-32B | 47.0 | - | 91.6 | 55.0 |
| DAPO-Qwen-32B [5] | 50.0 | - | - | - |
| DAPO-Qwen-32B[*] | 48.3 | 37.9 | 71.8 | 16.0 |
| Open-Reasoner-Zero-32B | 48.1 | 36.0 | 92.2 | 55.5 |

## 2.2 Key Design Principles

In this study, we explore best practices for reasoning-oriented RL training, emphasizing stability and scalability. Our key findings are summarized as follows:

**Choosing PPO over GRPO**   We select PPO over GRPO due to its superior value estimation enabled by a learned critic. This critic facilitates accurate token-level value estimation, effectively identifying and devaluing detrimental patterns such as repetitive behaviors, named credit assignment. Consequently, PPO achieves notably more robust advantage estimation compared to GRPO. Lacking a dedicated value network, GRPO struggles to distinguish genuinely correct responses from those occurring within negative patterns (*e.g.*, repetitive loops). This deficiency can misdirect reinforcement, leading to training instability and eventual collapse, an observation supported by community discussions[2]. Detailed analysis is provided in Section 3.3.

**Algorithm Implementations.**   Our empirical studies suggests that vanilla PPO already provides a highly stable and robust training across different model scales and training durations, without the need for additional algorithmic modifications. Nonetheless, appropriate implementations matter. Through extensive experiments, we found that the choice of GAE parameters substantially impacts performance in reasoning-oriented tasks. Specifically, the discount factor $\gamma$ controls the effective sequence length considered during training: a lower $\gamma$ assigns exponentially decreasing weights to future rewards, inducing the model to prematurely terminate generation in order to more immediately obtain rewards. On the other hand, the GAE parameter $\lambda$ balances bias and variance in advantage estimation. Crucially, in large-scale training scenarios, the substantial data volume naturally mitigates variance concerns, encouraging us to adopt a bias-free configuration. Consequently, by setting $\gamma = 1$ and $\lambda = 1$, we fully capture the long-term dependencies critical for reasoning tasks and achieve stable training. Fortuitously, this also leads to a significant simplification of the GAE advantage computation in our case:

$$\hat{A}_t^{GAE(\gamma=1,\lambda=1)} = R - V_\phi(s_t), \tag{4}$$

$$\mathcal{J}_{\text{value}}(\phi) = \frac{1}{2}\mathbb{E}_{\tau \sim \pi_{\theta_{\text{old}}}}\left[\sum_{t=0}^{T-1}(V_\phi(s_t) - R)^2\right], \tag{5}$$

where $R$ is the single terminal reward. Detailed derivation and pseudocode can be seen in appendix.

**Removing KL regularization.**   We achieve stable training without relying on any KL-based regularization techniques (*e.g.*, KL shaped rewards and loss), different from the de facto RLHF community [6] and Reasoner model [7, 2]. Intuitively, KL regularization constrains the policy model to remain close to the original base model distribution, potentially limiting exploration during policy optimization. By omitting KL regularization, our approach offers several practical advantages: (1) it obviates the need to navigate the large and challenging-to-tune design space inherent to KL regularization, greatly simplifying the training procedure; and (2) it lowers computational overhead and memory usage, eliminating the need to load the weight of a separate reference model and calculate log probabilities using it. Together, these benefits facilitate efficient and scalable large-scale RL training.

---

[2]OpenR1: discussion about vanilla GRPO reproduction link.

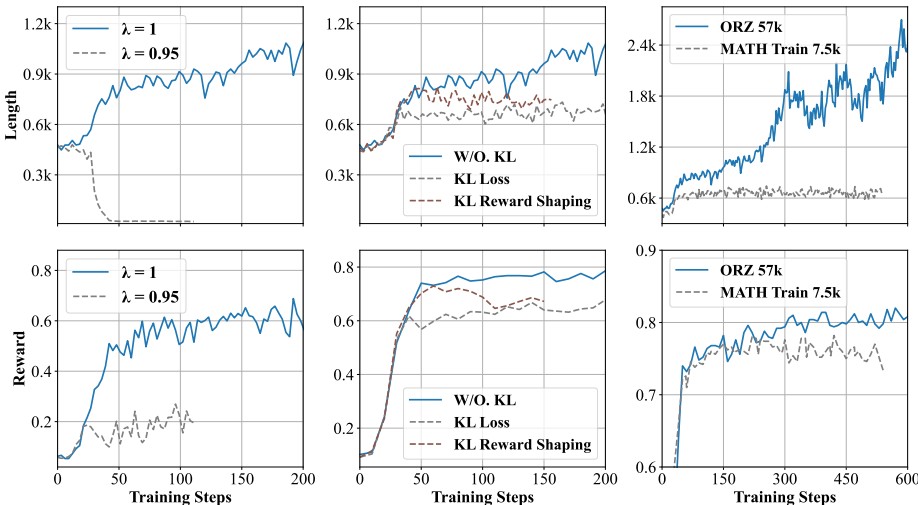

Figure 3: Ablation studies for key design choices in Open-Reasoning-Zero (ORZ). We use reward on training set or MATH500 as model performance metrics. **Left.** Comparison of different GAE $\lambda$ values. **Mid.** Comparisons of KL-related regularizations. **Right.** Data scale ablation study. These findings collectively inform our minimalist yet effective ORZ training recipe.

**Minimal Reward Function Design.**    In contrast to approaches such as DeepSeek R1, which utilize a dedicated format reward to enforce structured reasoning (e.g., enclosing thought processes within <think>...</think>), we demonstrate that the simplest, rule-based reward function is not only sufficient but also optimal, as minimal design leaves no room for potential reward hacking. Notably, even unaligned base models quickly adpot to desired format, suggesting this is a straightforward task without requiring complex reward engineering.

**Scale up Training Data.**    We identify that scaling up data quantity and diversity is pivotal for Reasoner-Zero training. While training on limited academic datasets like MATH train set leads to quick performance plateaus, our curated large-scale diverse dataset demonstrates impressive potential for continuous improvement without signs of saturation on both training and test sets.

## 2.3   Detailed Settings

We instantiate our ORZ approach by utilizing the Qwen2.5-{7B, 32B} base models as our main foundation. This methodology involves directly launching large-scale RL from these base models, bypassing any preliminary fine-tuning stages such as supervised fine-tuning (SFT) or distillation, a strategy also explored in recent works [8, 9]. Inspired by DeepSeek-R1-Zero [2], we design our prompt template to elicit the model to utilize inference computation, gradually mastering the reasoning ability, as shown in the Appendix. We detail our implementation below, focusing on the key components that enable effective and robust reasoning-oriented RL at scale.

**Data Curation.**    With careful consideration of scalability and robustness, our training data comprises tens of thousands of carefully curated question and answer pairs consisting of math and general reasoning tasks. Specifically, we curate our dataset through a comprehensive collection and cleaning process. First, we collect public data from various sources, including AIME (up to 2023), MATH [10], Numina-Math collection [11], Tulu3 MATH [12], OpenR1-Math-220k [13] and AoPS forum. We also synthesize general reasoning tasks using programmatic approaches as additional enrichment. These include logical puzzles, multi-step reasoning problems, and counterfactual scenarios that require the model to apply structured thinking across diverse domains. Considering the RL training paradigm's reliance on accurate reward signals, we exclude problems that are challenging to evaluate with our rule-based reward function, such as proof-oriented problems. This careful filtering ensures accurate and consistent reward computation during training, essential for stable policy optimization. We also employ LLM-based filtering to evaluate problem difficulty, removing samples with extreme pass rates to maintain a balanced dataset.

Table 2: Generalization performance of Open-Reasoner-Zero on MMLU and MMLU_PRO benchmarks. ORZ achieves superior performance on both benchmarks through RL training on reasoning tasks alone, surpassing Qwen2.5-Instruct without additional instruction tuning.

| Model | MMLU | MMLU_PRO |
|---|---|---|
| Qwen2.5-32B-Base | 83.3 | 55.1 |
| Qwen2.5-32B-Instruct | 83.2 | 69.2 |
| DAPO-Qwen-32B | 79.7 | 64.5 |
| Open-Reasoner-Zero-32B | **84.9** | **74.4** |

**Reward Function.** Unlike DeepSeek-R1-Zero [2], our scale-up RL training employs a minimalist rule-based reward function that solely checks answer correctness, without any additional format rewards. Specifically, this reward function is designed to extract the content between '<answer>' and '</answer>' tags during training and compare it with the reference answer. To maintain clarity and simplicity in scale-up RL, we implement a binary reward scheme - awarding a reward of 1 for exact matches with the reference answer, and 0 for all other cases. Surprisingly, we found that under our designed prompt even unaligned base model can yield well-formatted responses in high probability. Moreover, the base model can quickly learn the correct format and reinforce it for reasoning and answering incentivized by our simple rule-based reward function alone, as shown in Figure 4.

## 3 Experiments

In this section, we present comprehensive experimental results and analysis of our Open-Reasoner-Zero models. We begin with an in-depth analysis of training results and ablation studies. We then investigate the correctness and effectiveness of the value function. Finally, we discuss the evaluation results and in-depth analyze the training process. Hyperparameters are provided in the Appendix.

### 3.1 Training Results

We highlight key findings from our training experiments, examining performance through training reward, response lengths, and generation quality to provide a concise view of learning dynamics.

Figure 2 shows the training reward and average response length curves of our experiments for ORZ-{32, 7, 1.5, 0.5}B, where we observe consistent improvements in both metrics during training. This indicates that the models are effectively learning the desired reasoning behaviors.

To further understand the characteristics of the generated responses, Figure 4 (Right) illustrates the average length of all responses compared to the average length of responses that are correct and incorporate reflection steps. We identify five representative reflection patterns ('"wait,"', '"recheck"', '"retry"', '"alternatively,"', and '"however,"') and use this to determine whether a response is reflective, following a methodology similar to [14]. Notably, the average length of correct responses that utilize reflection is consistently greater than the overall average response length across all training steps. Furthermore, both of these length metrics exhibit a clear upward trend as training progresses.

### 3.2 Ablation Study

**GAE Analysis.** We investigated the impact of different GAE $\lambda$ values and found $\lambda$=1.0 to yield superior training stability and final performance. As shown in Figure 3 (Left), training with GAE $\lambda$=1.0 resulted in a reward that rapidly increases and then steadily grows, consistently outperforming $\lambda$=0.95, which exhibited much slower reward progression. In the Response Length, the GAE $\lambda$=1.0 curve maintains a reasonable increasing spead during the training process; while the GAE $\lambda$=0.95 leading to collapsed length dynamics. These findings indicate that GAE $\lambda$=1.0 can better balance the training stability and generation quality.

**KL Regularization Analysis.** We assessed the impact of KL Loss and Penalty on the performance and training dynamics of ORZ-7B. Figure 3 (Mid) clearly shows that omitting both KL Loss and Penalty (W/O. KL) achieve optimal training stability, performance, and response length scaling. Both

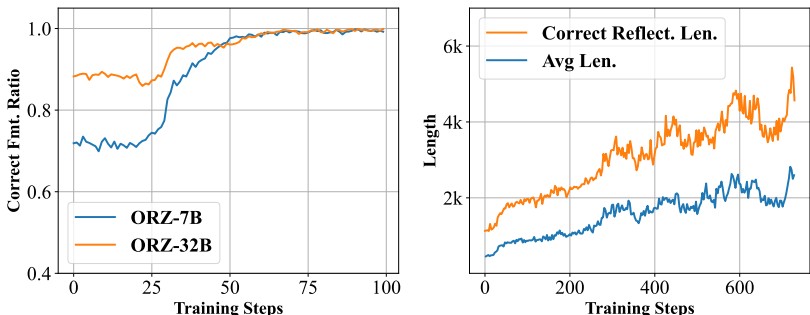

Figure 4: **Left.** Correct Format Ratio. Results demonstrate rapid adoption of structured reasoning patterns even by the base model trained on a simple outcome reward function, suggesting complex reward functions are unnecessary for Reasoner-Zero. **Right.** Reflection patterns in generation. Average Correct Reflection Length consistently exceeds Average Response Length during training.

KL Loss and KL Penalty mechanism not only slow down the training process but also consume additional computational resources. Furthermore, eliminating these components reduces hyperparameter tuning burden and implementation complexity, which is crucial for scaling up RL training effectively.

**Data Scale.**    We compared training with our ORZ 57k dataset against a classic academic dataset, MATH train 7.5k. As depicted in Figure 3 (Right), leveraging the larger ORZ 57k dataset leads to sustained improvements in both training reward and response length. In contrast, training with the smaller MATH train 7.5k dataset results in performance—both reward and length—plateauing early. These results underscore the pivotal role of data scale in enhancing training performance and affirm that increasing training data quantity can effectively improve the model's reasoning capabilities.

**Reasoner-Zero Training on Smaller Models.**    To demonstrate the robustness and versatility of our ORZ training methodology, we extend the same training pipeline to smaller-scale models, specifically Qwen2.5-{0.5,1.5}B. The evaluation results clearly indicate that our minimalist RL approach consistently improves reasoning capabilities even at substantially smaller model sizes. Remarkably, meaningful performance gains are observable even at the scale as small as 0.5B parameters. Detailed training curves for these smaller models are provided in the Appendix.

**Reasoner-Zero Training Across Diverse Domains.**    To test whether the ORZ recipe transfers beyond mathematics, we trained ORZ-7B in a mixed-domain setting (Math+Code+Puzzle+Instruction-Following at 40%:40%:15%:5%), while keeping all algorithmic settings unchanged. As summarized in Table 3, the ORZ-7B-Mixed-Domain model delivers large gains on non-math evaluations as well, without sacrificing mathematical competence. These results confirm that our ORZ recipe generalizes seamlessly across heterogeneous task distributions without requiring algorithmic modifications.

**Reasoner-Zero Training Across Different Model Families.**    We validate portability by applying the ORZ recipe to *Llama-3.1-8B Base* in the same mixed-domain setting, adjusting only the learning rate (from 1e-6 to 5e-7). Training remained stable with steady reward and response-length growth (see Figure 6). As detailed in Table 4, the resulting ORZ-Llama-3.1-8B model demonstrates substantial improvements over the base model. These results confirm that the ORZ methodology is architecture-agnostic and generalizes effectively beyond the Qwen model family.

**Training on Distillation Models.**    We further investigate preliminary results on applying the ORZ training pipeline to distilled models to enhance their reasoning capabilities. This two-stage approach follows DeepSeek-R1 [2]. Table 5 shows that our model yields essential further gains, with ORZ-R1-Distill-Qwen-14B outperforming larger distilled models like R1-Distill-Qwen-32B.

### 3.3   Analysis for Critic and Advantage Estimation

To further investigate the impact of our RL algorithm choices, particularly the preference for PPO over GRPO, we conducted detailed analyses of the learned value function (*i.e.*, critic), and its downstream

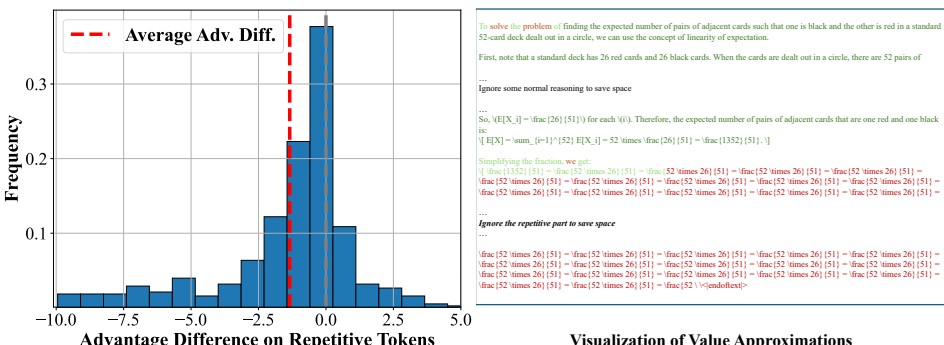

Figure 5: **Left.** Advantage comparison between PPO and GRPO on repetitive tokens. Our PPO are more negative advantages to repetitive patterns than GRPO, demonstrating superior penalization of undesirable. **Right.** Visualization of value approximations showing how $V_\phi(s_t)$ assigns lower values (red) to repetitive patterns and higher values (green) to coherent text, reflecting how the critic effectively identifies undesirable generation patterns.

effects. Our analyses reveal how the accurate critic influences advantage estimations and eventually translates into more effective policy updates compared to GRPO without critic.

Qualitatively, we observed that the value function $V_\phi(s_t)$ learned during PPO training effectively identifies repetitive patterns (*i.e.*, excessive repetition), which consistently occurs when a sudden collapse of vanilla GRPO as described in Section 2.2. As illustrated in Figure 5 (Right), states $s_t$ containing such repetitions are typically assigned lower values by $V_\phi$ (*i.e.*, lower expected future returns) compared to states with coherent patterns, a phenomenon known as credit assignment.

To quantify how this precise credit assignment benefits policy optimization, we performed a comparative analysis of advantage estimations between our PPO setup and a hypothetical GRPO setup, focusing on repetitive tokens. Inspired by Kimi k1.5[7], we first identified all tokens that appear after the onset of the first noteworthy repetitive pattern within a generation, which we designate as **tokens with repetitive patterns**. We then calculated the average advantage assigned to these specific tokens by our PPO GAE ($\lambda = 1, \gamma = 1$) setting (which includes batch-level advantage normalization). This was contrasted with the average advantage that GRPO would have assigned to the exact same tokens.

Figure 5 demonstrate during our ORZ-7B training, advantage assignments of our PPO configuration are consistently lower (*i.e.*, more negative) to these **tokens with repetitive patterns** compared to GRPO across the majority of training iterations. This finding reveals PPO's superior ability to penalize undesirable patterns, thereby fostering a more precise and robust learning signal that actively discourages degenerate outputs and promotes higher-quality responses, consistent with our empirical observations in large-scale RL. More detailed analyses are provided in the Appendix.

## 3.4 Evaluation Results

We present a comprehensive analysis of our results, demonstrating the effectiveness of ORZ across different model scales and benchmarks. Our experiments evaluate both training efficiency and reasoning performance, highlighting the scalability and generalization capabilities of our approach.

In our experiments, ORZ-32B demonstrates significant advancements in both efficiency and performance, as shown in Figure 1. The model achieves superior accuracies across all benchmarks, notably outperforming DeepSeek-R1-Zero-Qwen2.5-32B, while only requiring an order of magnitude fewer training steps. Moreover, we also compare ORZ-32B with DAPO-32B, another recent Reasoner-Zero model in Table 1. ORZ achieves comparable performance on AIME while requiring fewer the training iterations used by DAPO. Note that ORZ remarkably outperforms DAPO on MATH500 and GPQA Diamond. We observed that the released DAPO tends to answer with integer numbers (*e.g.*, "Answer: 2") even in multiple-choice questions without integers in the options. We hypothesize this behavior is related to their data curation and formatting approach, which transforms every answer into an integer for verification disambiguation. This limitation highlights the advantages of our data curation methodology, with high coverage and preprocessing to handle diverse answer formats correctly.

Table 3: The ORZ recipe transfers seamlessly to a mixed-domain setting without algorithmic changes. This approach yields substantial gains in coding (LCB), puzzle task (BBEH) and instruction following (IFEVAL) compared to the math-only training, yet no performance drop on math benchmarks.

| Model | AIME24 | GPQA Dia. | LCB | IFEVAL | BBEH |
|---|---|---|---|---|---|
| Qwen2.5-7B-Instruct | 10.1 | 35.7 | 17.1 | 55.8 | 10.3 |
| ORZ-7B | 17.9 | 36.6 | 2.3 | 37.6 | 4.0 |
| ORZ-7B-Mixed-Domain | **18.2** | **39.0** | **22.2** | **71.9** | **12.4** |

Table 4: The ORZ recipe generalizes robustly across different model families, delivering strong gains on diverse benchmarks when applied to Llama-3.1-8B Base model in the mixed-domain setting .

| Model | AIME24 | GPQA Dia. | IFEVAL | MMLU | MMLU_PRO |
|---|---|---|---|---|---|
| Llama-3.1-8B-Base | 0.16 | 14.29 | 15.96 | 27.83 | 10.56 |
| ORZ-Llama-3.1-8B | **0.31** | **27.27** | **68.67** | **60.02** | **33.51** |

We further illustrate the training dynamics of Open-Reasoner-Zero models across various sizes in Figure 2. Training Reward and Response Length demonstrate consistent and steady growth across all scales, highlighting the scalability of our minimalist reinforcement learning approach. Interestingly, the Response Length curve of the ORZ-32B model exhibits noticeable fluctuations, yet these fluctuations do not negatively impact training stability or the continuous growth of reward. This phenomenon indicates the robustness of our method against temporary variations in generated sequence lengths and motivates further investigation into understanding and leveraging this behavior.

Finally, we present the generalization capabilities of our models on comprehensive benchmarks like MMLU and MMLU_PRO. As shown in Table 2, ORZ-32B models demonstrate strong generalization capabilities, significantly outperforming Qwen2.5-Instruct-32B on MMLU, MMLU_PRO through pure scaled-up RL training on reasoning-oriented tasks, without any additional instruction tuning.

## 4   Related Work

**Scaling RL on Base Models for Reasoning**   The approach that applies RL directly to base models to master complex reasoning skills, referred to as Reasoner-Zero training, has gain far-reaching attention [2]. While several recent works [15, 16, 17] have proposed detailed training recipes for Reasoner-Zero approaches in pilot studies, ORZ stands as the first fully open-source implementation of large-scale reinforcement learning applied directly to base language models for reasoning.

Concurrent efforts have also explored Reasoner-Zero training at scale. DAPO [5] matches ORZ's AIME performance, but uses roughly fivefold more training iterations and underperforms on other benchmarks, potentially due to its data processing strategies. While VAPO [18] reports stronger AIME2024 accuracy with a similar iteration budget as DAPO, it scales less efficiently compared to ORZ, reaching only about 60% of ORZ's score at the same iteration budget. Notably, ORZ employs a simpler algorithm design avoiding the value function learning challenges faced by others, establishing it as a more effective and accessible baseline for future research.

**Scaling RL on Reasoning-Enhanced Models**   Another important line of work [2, 8, 19, 20] endeavors to scale up RL training on reasoning-enhanced Models. Theses models typically first acquire advanced reasoning patterns from existing reasoning models or high-quality human-labeled data [21, 22, 23, 24, 25] through techniques such as SFT distillation or other cold-start approaches. A key advantage of this pre-instruction is that the subsequent RL training is more stablem and these models can achieve superior performance under the same RL compute budget compared to the Reasoner-Zero training. We validate that ORZ recipe is also highly effective for such distilled models.

**Our Contributions**   In contrast to previous work, ORZ offers the most comprehensive open framework for Reasoner-Zero training. Our main contributions include: (1) a simple yet scalable RL algorithm implementation that serves as a strong and accessible baseline for future research; (2) extensive configurations and benchmarks spanning models from 0.5B to 32B parameters; (3) the

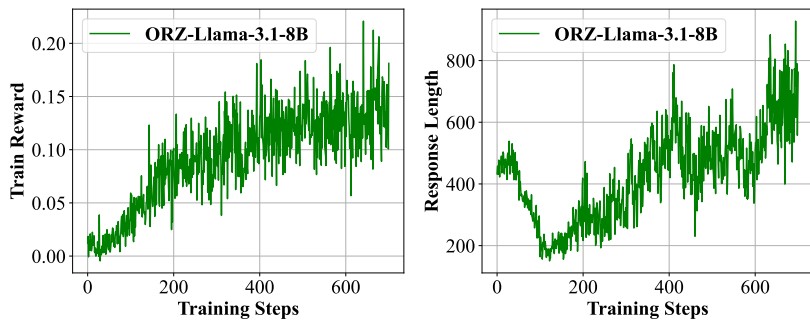

Figure 6: Train-time scale up on Train Reward and Response Length of ORZ-Llama-3.1-8B in the mixed-domain setting. The application of the ORZ recipe exhibits stable training dynamics on a different model architecture, confirming its robustness and model-family-agnostic nature.

Table 5: The ORZ training recipe enables the DeepSeek-R1-Distill-Qwen-14B model to grasp advanced reasoning patterns distilled from stronger reasoning models, substantially boosting its performance. This ORZ-R1-Distill-Qwen-14B achieves strong results on reasoning benchmarks, even surpassing the larger DeepSeek-R1-Distill-Qwen-32B model.

| Model | AIME2024 | AIME2025 | MATH500 | GPQA Dia. |
|---|---|---|---|---|
| DeepSeek-R1-Distill-Qwen-14B | 69.7 | 49.1 | 93.9 | 59.1 |
| DeepSeek-R1-Distill-Qwen-32B | 72.6 | **60.0** | 94.3 | **62.1** |
| ORZ-R1-Distill-Qwen-14B | **75.2** | **60.0** | **95.6** | 60.4 |

largest verified reasoning dataset; and (4) state-of-the-art training efficiency with minimal iterations. Together, these efforts establish a foundational open framework for large-scale RL research on LLMs, enabling broader community participation in advancing reasoning capabilities.

# 5   Limitations

While ORZ demonstrates significant advancements in scaling reasoning-oriented RL training, we acknowledge certain limitations in the current work. Firstly, our investigation primarily focuses on mathematical and general reasoning tasks. We have not included results on code generation or other programming-related reasoning tasks. Exploring the efficacy of ORZ on code-based benchmarks, or investigating the potential benefits and challenges of combining mathematical and code reasoning within a unified training framework, remains a critical aspect of developing comprehensive reasoning agents. Secondly, while we successfully implement training time scaling in RL, we have yet to fully explore test time scaling like OpenAI o1. Future work should investigate scaling test time computation through multi-turn interactions for contextual reasoning, value models for trajectory assessment, and multi-agent scenarios for developing more sophisticated reasoning strategies.

# 6   Conclusion and Discussions

We present Open-Reasoner-Zero (ORZ), the first comprehensive open-source implementation of large-scale reasoning-oriented RL training. Our experiments show that vanilla PPO with GAE ($\lambda = 1$, $\gamma = 1$) and simple rule-based rewards, without KL regularization, effectively scales reasoning capabilities in language models. This minimalist approach achieves competitive results compared to DeepSeek-R1-Zero while using significantly fewer training iterations. Our work provides in-depth analysis on training dynamics, model behaviors, and advantage estimation. These insights offer practical guidance on scaling RL for complex reasoning tasks, addressing common challenges in stability and convergence. By releasing our complete training resources—code, configurations, data, and model weights across various sizes—we aim to democratize access to reasoning-oriented RL and provide valuable insights for the community.

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

In this Supplementary Material, we provide more elaboration on the implementation details, experiment results, and qualitative results. Specifically, we present more evaluation results in Section A, thorough implementations of the model training in Section B, and additional analyses and experiments in Section C and D. These materials offer deeper insights into our methodology, experimental validation, and qualitative findings that support the conclusions presented in the main text.

## A    More Evaluation Results

In this section, we provide detailed results from evaluating ORZ models of varying parameter counts (0.5B, 1.5B, 7B, and 32B) across multiple reasoning-oriented benchmarks. Specifically, we report performance on AIME 2024, AIME 2025, MATH500, and GPQA Diamond. The results (see Table 6) clearly demonstrate consistent improvements in reasoning ability with increased model size, underscoring the strong scaling properties of our minimalist RL setup. We release these comprehensive evaluation results as a reference to facilitate further research and reproducibility. We also provide the training performance curves for ORZ-{0.5B, 1.5B} in Figure 7.

Table 6: Reasoning-oriented benchmark performance across Open-Reasoner-Zero model sizes.

| Model | AIME 2024 | AIME 2025 | MATH500 | GPQA Dia. |
|---|---|---|---|---|
| ORZ-0.5B | 1.0 | 0.2 | 31.0 | 12.1 |
| ORZ-1.5B | 3.5 | 1.0 | 58.0 | 16.8 |
| ORZ-7B | 17.9 | 15.6 | 81.4 | 36.6 |
| ORZ-32B | 48.1 | 36.0 | 92.2 | 55.5 |

## B    Detailed Setting for Training

We initialize both our policy and critic networks with Qwen-2.5 base models (7B and 32B variants), where value head is randomly initialized from $\mathcal{U}(-\sqrt{5}, \sqrt{5})$ with no bias term. The policy and critic do not share weights during training. For both policy and critic networks, we employ AdamW optimizer with $\beta = [0.9, 0.95]$ without weight decay. The learning rates are set to $1 \times 10^{-6}$ and $5 \times 10^{-6}$ for the policy and critic networks, respectively. The learning rate schedulers are both constant learning rate with linear warm-up of 50 optimizer steps. We employ sample packing during training. Prompt for ORZ model training and evaluation are provided in Table 7.

---

A conversation between User and Assistant. The user asks a question, and the Assistant solves it.
The assistant first thinks about the reasoning process in the mind and then provides the user
with the answer. The reasoning process and answer are enclosed within <think> </think> and
<answer> </answer> tags, respectively, i.e., <think> reasoning process here </think>
<answer> answer here </answer>. User: You must put your answer inside <answer> </answer> tags, i.e.,
<answer> answer here </answer>. And your final answer will be extracted automatically by the \boxed{} tag.
{{prompt}}
Assistant: <think>

---

Table 7: Template for Open-Reasoner-Zero. prompt will be replaced with the specific reasoning question during generation.

Each generation step contains 128 unique prompts sampled from the dataset, and policy generates 64 responses per prompt with temperature and top-p both set to 1.0. To maintain training stability, we implement strict on-policy optimization for the policy network, where each generation corresponds to exactly one policy optimization step. The critic network, being less sensitive to off-policy updates, processes the experiences in 12 mini-batches, effectively performing 12 optimization steps per iteration. We apply batch level advantage normalization in the training. Notably, our training process operates stably without any KL-related regularization terms or entropy bonuses, demonstrating that vanilla PPO can achieve stable training without these commonly used stabilization techniques.

For the 32B variant, we introduce an additional "annealing" training stage inspired by analogous practices in large language model pre-training [26]. Specifically, we leverage the training process of

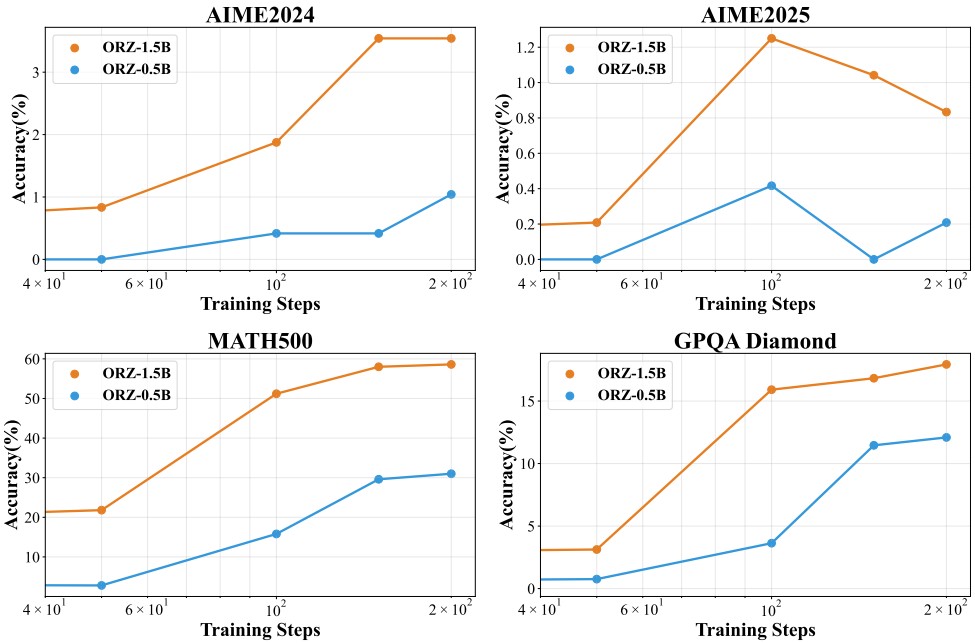

Figure 7: Evaluation performance of Open-Reasoner-Zero-{0.5B, 1.5B}. We report the average accuracy on the benchmark dataset for each question with 16 responses.

our 32B model itself to identify challenging and high-quality prompts for this annealing stage. We pinpoint 13k particularly difficult prompts, defined as those where the model achieves fewer than 4 correct answers out of a total of 64 attempts during the first 1100 steps of training. These identified prompts are then selectively used in a final training stage of 100 additional steps and apply a linear learning rate decay schedule, reducing to $3 \times 10^{-7}$. This targeted training phase is explicitly designed to enhance the model's capability on more complex reasoning tasks.

For the training of the ORZ-R1-Distill-Qwen-14B model, we initialized its weights from the DeepSeek-R1-Distill-Qwen-14B model. We utilize the mined 13k difficult prompts as training data. All other hyperparameters follow the basic configuration of the ORZ model family. The reported results correspond to the checkpoint at 300 training iterations.

## C  Additional Analyses and Experiments

### C.1  More Analysis for Critic and Advantage Estimation

As noted in the main text, vanilla GRPO often suffers from significant training instability, a phenomenon also observed in many community implementations. This instability typically manifests as a deterioration in generation quality midway through training, with models tending to produce repetitive or incoherent text. Our large-scale experimental validation strongly corroborates these observations and highlights the superior training stability of PPO compared to GRPO, as shown in Figure 8.

### C.2  Ablation on Data Curation

Based on our analysis of data quality issues, we conduct comprehensive ablation studies to evaluate how different data curation strategies affect model training stability and performance. Motivated by OpenR1's finding [13] that SFT performance degradation on Chinese subsets was due to simpler question patterns, we experiment with two data curation approaches: using English-only data versus using both English and Chinese data. As shown in Figure 9, the English-only dataset yields superior training stability and final model performance.

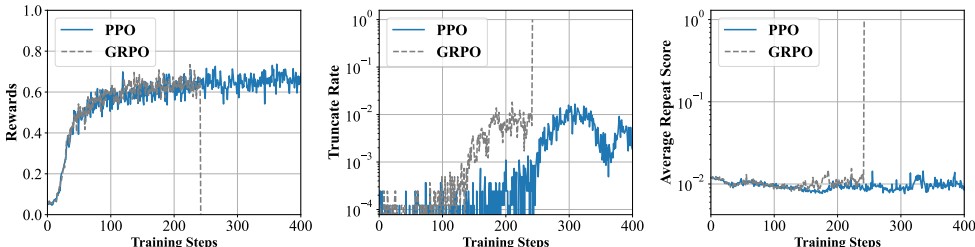

Figure 8: PPO vs. GRPO stability in the ORZ-7B setting. GRPO empirically demonstrates severe training instability around 240 training steps: its reward suddenly destabilizes, and responses degenerate as both Truncate Rate and Average Repeat Score surge to 1.0. In contrast, PPO maintains stable rewards and low Truncate/Repeat Scores throughout.

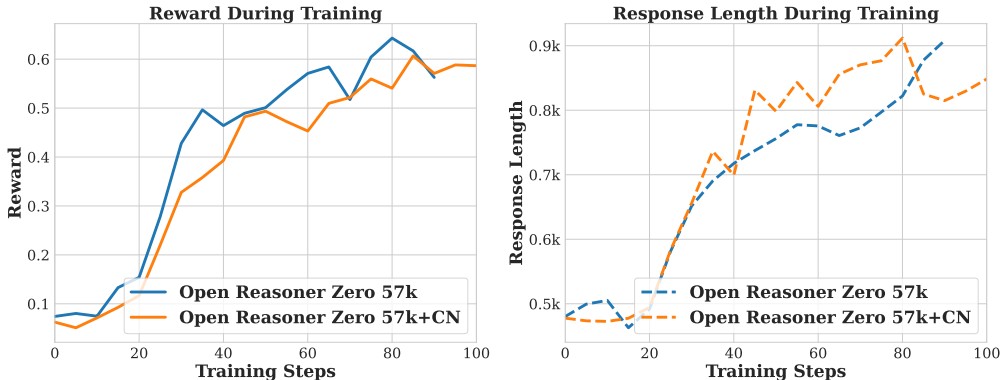

Figure 9: Data Curation Ablation. CN represents Chinese data and EN represents English data. English-only dataset yields superior training stability and final model performance.

## D   Derivation and Code for PPO with GAE($\gamma = 1, \lambda = 1$)

This section provides a detailed derivation for the GAE in the specific case where $\gamma = 1$ and $\lambda = 1$.

We substitute $\gamma = 1$ and $\lambda = 1$ into the general GAE formulation (defined in the main text). Recall that $\delta_{t+k} = r_{t+k} + \gamma V_\phi(s_{t+k+1}) - V_\phi(s_{t+k})$. With $\gamma = 1$, this becomes $\delta_{t+k} = r_{t+k} + V_\phi(s_{t+k+1}) - V_\phi(s_{t+k})$. Thus:

$$
\begin{aligned}
\hat{A}_t^{GAE(\gamma=1,\lambda=1)} &= \sum_{k=0}^{T-t-1} (1 \cdot 1)^k \delta_{t+k} \\
&= \sum_{k=0}^{T-t-1} \left( r_{t+k} + V_\phi(s_{t+k+1}) - V_\phi(s_{t+k}) \right) \\
&= \sum_{k=0}^{T-t-1} r_{t+k} + \sum_{k=0}^{T-t-1} \left( V_\phi(s_{t+k+1}) - V_\phi(s_{t+k}) \right) \quad (6) \\
&= R - V_\phi(s_t) \quad (7)
\end{aligned}
$$

The step from (6) to (7) follows because: (i) the sum of rewards $\sum_{k=0}^{T-t-1} r_{t+k}$ equals the single terminal reward $R$ for the trajectory, as intermediate rewards are zero; and (ii) the second sum $\sum_{k=0}^{T-t-1} \left( V_\phi(s_{t+k+1}) - V_\phi(s_{t+k}) \right)$ is a telescoping series that evaluates to $V_\phi(s_T) - V_\phi(s_t)$, where $s_T$ is the terminal state, and $V_\phi(s_T) = 0$.

Now we derive the simplified form for the value target $V_t^{\text{target}}$. As defined in the main text, we have:

$$
\begin{aligned}
V_t^{\text{target}} &= \hat{A}_t^{GAE(1,1)} + V_\phi(s_t) \\
&= (R - V_\phi(s_t)) + V_\phi(s_t) \\
&= R
\end{aligned}
\tag{8}
$$

Substituting this simplified target $V_t^{\text{target}} = R$ into the general value loss formulation (defined in the main text):

$$
\mathcal{J}_{\text{value}}(\phi) = \frac{1}{2}\mathbb{E}_{\tau \sim \pi_{\theta_{\text{old}}}} \left[ \sum_{t=0}^{T-1} (V_\phi(s_t) - R)^2 \right]
\tag{9}
$$

We also provide a detailed algorithm implementation in 1.

---

**Algorithm 1** PPO with GAE($\gamma = 1, \lambda = 1$)

---

**Require:** Initial policy parameters $\theta_0$, initial value parameters $\phi_0$, prompt dataset $\mathcal{D}$.
**Require:** Hyperparameters: clip range $\epsilon$, trajectories per prompt $n$, minibatch size $M$.
 1: Initialize policy $\pi_\theta \leftarrow \pi_{\theta_0}$, value function $V_\phi \leftarrow V_{\phi_0}$.
 2: Initialize $\theta_{\text{old}} \leftarrow \theta_0$, $\phi_{\text{old}} \leftarrow \phi_0$.
 3: **for** iteration $= 1, 2, ...$ **do**
 4:     Initialize experience buffer $\mathcal{B} \leftarrow \emptyset$.
 5:                                                        ▷ — Rollout Phase —
 6:     Sample batch of prompts $\{q_j\}$ from $\mathcal{D}$.
 7:     **for all** prompts $q_j$ in the batch **do**
 8:         Generate trajectory $\tau = (s_0, a_0, ..., s_{T-1}, a_{T-1})$ using policy $\pi_{\theta_{\text{old}}}$.
 9:         Compute terminal reward $R \in \{0, 1\}$ for $\tau$.
10:         Compute value estimate $V_t^{\text{old}} = V_{\phi_{\text{old}}}(s_t)$.
11:         Compute advantage $\hat{A}_t = R - V_t^{\text{old}}$.            ▷ Using $\gamma = 1, \lambda = 1$
12:         Store tuple $(\tau, \log \pi_{\theta_{\text{old}}}(a_t|s_t), R, \hat{A}_t)$ in buffer $\mathcal{B}$.
13:     **end for**
14:                                                    ▷ — Update Phase —
15:                                                    ▷ Update critic model
16:     **for all** minibatches $(\tau, R)$ from $\mathcal{B}$ **do**
17:         Compute current critic value $V_\phi(s_t)$.
18:         $L_{\text{VF}}(\phi) = \frac{1}{2}(V_\phi(s) - R)^2$.
19:         Backward and update $\phi$
20:     **end for**
21:                                                    ▷ Update policy model
22:     **for all** minibatches $(\tau, \log \pi_{\text{old}}(a|s), R, \hat{A})$ from $\mathcal{B}$ **do**
23:         Compute current policy log-probability $\log \pi_\theta(a|s)$.
24:         Calculate probability ratio $\rho(\theta) = \exp(\log \pi_\theta(a|s) - \log \pi_{\text{old}}(a|s))$.
25:         $L_{\text{CLIP}}(\theta) = \min\left(\rho(\theta)\hat{A}, \text{clip}(\rho(\theta), 1 - \epsilon, 1 + \epsilon)\hat{A}\right)$.
26:         Backward and update $\theta$
27:     **end for**
28:     Update old parameters: $\theta_{\text{old}} \leftarrow \theta$, $\phi_{\text{old}} \leftarrow \phi$.
29: **end for**
**Ensure:** Final policy parameters $\theta$.

---

