# OpenReview forum: "Open-Reasoner-Zero: An Open Source Approach to Scaling Up Reinforcement Learning on the Base Model"
_NeurIPS.cc/2025/Conference — NeurIPS 2025 poster_

### Official Review · Reviewer_Bk7x · 2025-06-26

**Clarity:** 3
**Significance:** 4
**Originality:** 3
**Rating:** 5
**Confidence:** 4

**Summary:**

This paper proposes an open-source framework that enables DeepSeek-R1-Zero-style scaling in both performance and response length, using PPO combined with a simple exact-matching rule-based reward. The method involves PPO with multiple rollouts per prompt, no KL regularization, careful data curation, and GAE with γ=1 and λ=1 (collapses to the Monte-Carlo return minus value baseline). Experiments on mathematical reasoning tasks demonstrate that this PPO setup achieves efficient and stable training dynamics, outperforming GRPO and DAPO baselines.

**Questions:**

1. The increase in response length and reward does not necessarily indicate improved reasoning ability, it may also result from reward hacking. How do the authors rule out this possibility?
2. In Fig.1 and Fig.2, the training steps appears to scale with model size. What is the rationale behind this? What would happen if training continued further?
3. Why does ORZ-32B exhibit noticeable fluctuations in response length (Fig.2), while smaller models (e.g., ORZ-7B, ORZ-1.5B, ORZ-0.5B) do not?
4. How is the threshold between low-value and high-value regions defined (Fig.5)?

**Ethical Concerns:**

["NO or VERY MINOR ethics concerns only"]

**Final Justification:**

My concerns have been addressed. I will keep my original score, which supports acceptance.

**Limitations:**

Yes.

**Paper Formatting Concerns:**

No formatting issue observed.

**Quality:**

4

**Strengths And Weaknesses:**

### Strength
This work introduces a valuable framework with insightful empirical observations, which could be impactful for the LLM reasoning and RL post-training communities. The analysis of the critic and advantage estimation is particularly helpful, showing the clear benefits of using a learnable critic.

### Weakness
The comparisons and analysis of ORZ primarily focus on performance and response length curves. However, merely presenting the curves does not rule out the influence of reward hacking, especially after the KL penalty has been removed. Providing additional qualitative comparisons of the responses can make the paper more convincing.

---

> ### Author Rebuttal · Authors · 2025-07-31
>
> We sincerely thank the reviewer for the positive assessment and thoughtful questions about our work. We appreciate the recognition of our framework's value and the insightful analysis on critic and advantage estimation. We address each concern below.
> ### On Reward Hacking Concerns
> We appreciate the reviewer's concern about potential reward hacking. However, our experimental setup has taken this into consideration, and greatly reduce the potential space for reward hacking, fundamentally different from RLHF settings where this issue commonly arises:
> - We focus exclusively on reasoning tasks with verifiable ground truth answers
> - Our reward function is a simple exact-match criterion against these ground truths
> - Unlike RLHF with learned reward models that can be exploited, our binary exact-match reward cannot be "gamed" through superficial response modifications
> The traditional reward hacking in RLHF manifests when models optimize for high reward model scores while producing qualitatively worse outputs. In our setting, achieving higher rewards directly corresponds to solving more problems correctly.
>
> ### Evidence of Genuine Reasoning Improvement
> Beyond the reward and response length curves, we want to share more evidence that ORZ develops genuine reasoning capabilities. First, the model shows consistent performance improvements across a wide range of held-out evaluation sets, indicating true generalized intelligence improvement. We also conducted careful manual inspection of model outputs throughout training. Here's a representative example from ORZ-32B:
> - Question: Find the greatest integer value of $b$ for which the expression $\frac{9x^3+4x^2+11x+7}{x^2+bx+8}$ has a domain of all real numbers.
> - ORZ-32B's approach (summarized): The model correctly identifies that the domain constraint requires the denominator to never equal zero → formulates this as requiring the discriminant $b^2-32<0$ → solves to get $|b|<\sqrt{32}=4\sqrt{2}\approx 5.656$ → systematically checks boundary cases to conclude $b=5$ is the maximum integer value → performs multiple rounds of self-reflection to verify the answer.
>
> This example illustrates the model's ability to: (i) correctly interpret mathematical constraints, (ii) apply appropriate algebraic techniques, (iii) perform systematic case analysis, and (iv) engage in self-verification. These are hallmarks of genuine mathematical reasoning, not reward exploitation. We will include additional qualitative examples and more detailed analysis in the revised version.
>
> ### Training Duration for Different Model Scales
> The reviewer correctly observes that training steps are different across model sizes. We clarify that this was not a deliberate design choice but rather a practical constraint: experiments on smaller scale models are not of top priority in our plan, and it gets terminated early due to limited computation and time constraints. We are currently resuming experiments to explore longer training horizons for these smaller models and will update the paper with extended training curves.
>
> ### Response Length Fluctuations in ORZ-32B
> This is indeed an interesting phenomenon that we've investigated further since submission. Our additional experiments reveal that learning rate tuning is the key factor:
> - When we reduced the 32B model's learning rate from 1e-6 to 5e-7, length growth became much more stable.
> - Conversely, increasing the 7B model's learning rate from 1e-6 to 2e-6 induced similar fluctuations.
> We will include these findings in the appendix of the revised version.
>
> ### Clarification on Figure 5 Visualization
> We apologize for any confusion in the figure caption. We actually use a continuous color map rather than discrete thresholds in text visualization: values closer to 0 appear redder, and values closer to 1 appear greener. And as you can see in the original submission's Figure 5, the critic learns to assign lower values to repetitive/degenerate text and higher values to meaningful reasoning steps. We will revise the caption to make this color mapping logic clearer.

---

> > ### Comment · Reviewer_Bk7x · 2025-08-04
> >
> > The author's response has addressed most of my concerns. I will keep my original score.

---

> > > ### Author Response · Authors · 2025-08-04
> > >
> > > Thank you for taking the time to read our response and for your thoughtful feedback throughout the review process!
> > > We’re glad our clarifications were helpful, and we appreciate your comments, which have strengthened the paper.

---

### Official Review · Reviewer_2p6y · 2025-06-29

**Clarity:** 4
**Significance:** 2
**Originality:** 1
**Rating:** 4
**Confidence:** 5

**Summary:**

This paper presents Open-Reasoner-Zero, an open-source implementation of large-scale reasoning-oriented reinforcement learning (RL) directly on base LLMs. The work advocates for a minimalist pipeline based on vanilla PPO with simple rule-based rewards and demonstrates scalability across models (0.5B to 32B) and benchmarks (AIME, MATH500, GPQA). The authors compare their method against DeepSeek-R1-Zero and QwQ models, claiming superior performance with significantly fewer training steps.

**Questions:**

1. Can you provide results or qualitative insights on whether similar improvements are seen when applying your pipeline to non-Qwen base models?

2. Have you tested or considered other RL algorithms to verify if PPO is indeed optimal for this setting?

3. What is the expected failure mode or limitation of this minimalist RL approach when scaling to other domains or tasks beyond reasoning?

**Ethical Concerns:**

["NO or VERY MINOR ethics concerns only"]

**Final Justification:**

The response from the author has partially address my previous concerns (e.g., evaluation on non-Qwen models and coding benchmark, discussion on other RL approach). Even though the novelty for the broader LLM research might be limited, the contribution of open sourcing outweighs such concern, and I would adjust the rating based on these considerations.

**Limitations:**

yes

**Quality:**

3

**Strengths And Weaknesses:**

Strengths:

1. Open-source contribution: The release of code, weights, and data contributes to the reproducibility and accessibility of RL for LLMs, enabling further experimentation and community participation.

2. Empirical findings on minimalist design: The authors provide insightful observations that vanilla PPO without KL regularization and with basic reward functions can scale well, which may simplify the design of future RL pipelines for LLMs.

3. Efficient performance scaling: The reported results show strong gains in benchmark performance and response length with significantly fewer steps, indicating the practical value of the proposed pipeline.

Weaknesses:

1. Limited novelty in methodology: The paper primarily repackages existing techniques (e.g., vanilla PPO, GAE, rule-based rewards) and applies them to a specific training setting. There is no substantive methodological innovation in RL algorithms, reward modeling, or data curation that would be considered novel in the NeurIPS community.

2. Evaluation restricted to Qwen-based models: The experiments focus solely on Qwen2.5 models. It remains unclear whether the observed benefits and scalability extend to other foundation models (e.g., LLaMA, Mistral, or GPT-like architectures). Broader evaluation would better support the claimed generality of the pipeline.

3. No comparison of alternative RL methods: PPO is the sole algorithm evaluated. The authors do not justify why PPO is preferable over other modern RL methods (e.g., TRPO, REINFORCE++, DPO, P3O). It would strengthen the contribution to examine whether PPO’s stability is unique or if similar trends arise with other approaches.

---

> ### Author Rebuttal · Authors · 2025-07-31
>
> Thanks to the reviewer for the detailed assessment and for flagging areas where broader evidence would help.
> ### Novelty
> While the core algorithm is purposefully minimal, the contribution is the first fully open and efficient reproduction of DeepSeek-R1-Zero. Matching its accuracy with 10x fewer steps, dissecting which components are dispensable, and providing a reusable training harness help the field move beyond opaque, resource-intensive recipes.
> Our pipeline, while retaining the simplicity of vanilla PPO, introduces three scaling-critical components that depart markedly from originally widely-adpoted PPO implementations—and these deviations are uniquely critical for large-scale reasoning-oriented RL.
>
> ### Extension to domains beyond math and reasoning
> Our recipe transfers to new domains (e.g. code, puzzle, instruction-following dialogues) without any modifications. Note that the original submission has already included puzzle-style tasks such as table-join, table-reformat, zebra, and connections alongside math problems. To further demonstrate generalizability, we conducted additional experiments on Qwen-2.5-7B base model with a mixed-domain training distribution (math:code:puzzle:instruction-following = 40%:40%:15%:5%) using the identical ORZ recipe. Training progressed smoothly with consistent increases in both reward and response length. The table below compare benchmark performance between this mixed-domain model and our math-focused model across multiple evaluations, where the final column NAIVE_AVG is benchmark level arithmetic average for all listed benchmarks.
> | init_model | training domain | train_iter | AIME24 | AIME25 | MATH500 | GPQA_DIA | LCB_V5 | LCB_V6 | IFEVAL | LIVEBENCH | MMLU_TEST | MMLU_PRO | NAIVE_AVG |
> |------------|-----------------|------------|--------|--------|---------|----------|--------|--------|--------|-----------|-----------|----------|-----------|
> | Qwen2.5-7B-Instruct | - | - | 10.10% | 6.04% | 73.80% | 35.70% | 12.70% | 17.14% | 55.82% | 28.83% | 72.66% | 50.91% | 36.37% |
> | Qwen2.5-7B-Base | Math+Code+Puzzle+IF | 500 | 16.46% | 14.69% | 79.00% | 35.89% | 20.79% | 23.14% | 66.27% | 40.51% | 73.81% | 54.06% | 42.46% |
> | Qwen2.5-7B-Base | Math | 500 | 16.20% | 13.13% | 79.60% | 36.71% | 1.30% | 1.72% | 40.27% | 33.99% | 74.67% | 56.08% | 35.37% |
> | Qwen2.5-7B-Base | Math | 700 | 17.50% | 15.00% | 81.00% | 37.85% | 2.15% | 2.29% | 37.62% | 35.97% | 75.04% | 57.80% | 36.22% |
>
> The enlarged task mix delivers substantial gains beyond pure mathematics: IFEVAL, LIVEBENCH, and the logic/code-heavy LCB suites improve sharply, yet the minimalist PPO recipe remains untouched, clear evidence of stronger reasoning transfer.
> This confirms that our recipe generalizes seamlessly to diverse domains without any algorithmic changes.
>
> ### Generalization to non-Qwen model families
> We successfully applied our ORZ recipe to Llama-3.1-8B base model with minimal adaptation (in the above mentioned mixed domain training setting, only adjusting the learning rate from 1e-6 to 5e-7). Training metrics show stable progression with both reward and response length increasing steadily throughout training, as shown in the table below:
> | train_iters | 0 | 100 | 200 | 300 | 400 | 500 |
> |-------------|---|-----|-----|-----|-----|-----|
> | response_len | 458 | 230 | 307 | 356 | 546 | 496 |
> | reward | 1.10% | 3.49% | 8.05% | 9.20% | 11.91% | 12.31% |
>
> Offline evaluations also show significant performance improvements across benchmarks as table below. The ORZ-Llama3.1-8B model in the table below corresponds to the checkpoint at iteration 500 (iter500) from applying our ORZ recipe to the Llama3.1-8B-Base model. The NAIVE_AVG is benchmark level of arithmetic mean of all listed benchmarks.
> | model | AIME24 | AIME25 | MATH500 | GPQA_DIA | IFEVAL | LIVEBENCH | MMLU_TEST | NAIVE_AVG |
> |-------|--------|--------|---------|----------|--------|-----------|-----------|-----------|
> | Llama-3.1-8B-Instruct | 2.71% | 0.42% | 39.60% | 27.68% | 54.21% | 19.36% | 62.26% | 29.46% |
> | Llama3.1-8B-Base | 0.16% | 0.00% | 0.29% | 14.29% | 15.96% | 3.56% | 27.83% | 8.87% |
> | ORZ-Llama3.1-8B | 0.52% | 0.16% | 21.93% | 26.70% | 66.96% | 27.17% | 61.21% | 29.24% |
>
> ### Alternative RL algorithms.
> We have already benchmarked against GRPO—the most widely used variant in this setting—and found PPO to be both more sample-efficient and more stable. TRPO is seldom applied to LLMs because its second-order updates are computationally expensive, while REINFORCE++, a critic-free method akin to GRPO, shows similar convergence behaviors without offering clear advantages.
>
> We hope these additions clarify the generality of the approach, justify our focus on PPO, and acknowledge its current limits.

---

### Official Review · Reviewer_wfie · 2025-07-02

**Clarity:** 4
**Significance:** 3
**Originality:** 3
**Rating:** 5
**Confidence:** 4

**Summary:**

The paper introduces Open-Reasoner-Zero-32B (ORZ-32B), an open-source large-scale reasoning model trained with PPO and Generalized Advantage Estimation (GAE) using Qwen 2.5-32B base model. By dissecting the contribution of individual algorithmic components and their hyperparameters, the authors show that vanilla PPO, when paired with their recommended hyperparameter settings, outperforms the GRPO variant of Guo et al. (2025). Despite operating with only 10 % of DeepSeek-R1-Zero’s total training budget, ORZ-32B achieves higher average scores on a suite of mathematical and general-reasoning benchmarks.

Daya Guo, Dejian Yang, Haowei Zhang, Junxiao Song, Ruoyu Zhang, Runxin Xu, Qihao Zhu, Shirong Ma, Peiyi Wang, Xiao Bi ,et al. Deepseek-r1: Incentivizing reasoning capability in llms via reinforcement learning. arXiv preprint arXiv:2501.12948,2025.

**Questions:**

1. Beyond benchmark accuracy, does ORZ-32B exhibit issues such as poor readability, mixed-language outputs, or over-compression?
2. Have you identified any persistent weaknesses (e.g., hallucination on domain-specific tasks, sensitivity to prompt length) in ORZ-32B?

**Ethical Concerns:**

["NO or VERY MINOR ethics concerns only"]

**Limitations:**

The authors have adequately addressed tthe limitations of their work.

**Paper Formatting Concerns:**

I have no paper formatting concerns.

**Quality:**

3

**Strengths And Weaknesses:**

### Strengths
- Thorough experimental design with ablation studies that isolate the impact of each component.
- Transparent reporting of hyperparameters, training budget, and evaluation setup.
- Writing is concise and well structured; figures and tables are easy to follow.
- Provides the first comprehensive head-to-head comparison of vanilla PPO versus GRPO in this setting.
### Weaknesses
-  Limited analysis of downstream failure modes or undesirable behaviors beyond benchmark scores.
- The proposed ORZ pipeline resembes prior PPO-finetuning pipelines; novelty lies more in systematic analysis than in algorithmic invention.

---

> ### Author Rebuttal · Authors · 2025-07-31
>
> Thanks to the reviewer for the encouraging evaluation and for highlighting areas where a deeper analysis could strengthen the paper. We respond to each point below.
>
> ### General-scene quality
> To go beyond reasoning benchmark scores and examine the quality of answer part (not the thinking part) of model output, we now include results on ArenaHard v2, an open-ended dialogue benchmark that exhibits strong correlation and separability with LMArena “judge-vote” rankings. Notably, ORZ-32B outperforms the strong Qwen-2.5-32B-Instruct baseline on ArenaHard v2—by +0.7 % in overall win rate (6.50 % vs 5.81 %) and an even larger +3.4 % in style-control win rate (11.40% vs 7.96%). These margins confirm that ORZ-32B generates generally fluent, well-controlled dialogue without mixed-language artifacts or over-compression in the answer part of model's output.
> | model | win_rate | style_control_win_rate |
> |-------|----------|------------------------|
> | Qwen2.5-32B-Instruct | 5.81% | 7.96% |
> | ORZ-32B | 6.50% | 11.40% |
>
> However, after careful investigation, we find that ORZ-32B develops highly flexible patterns in the thinking part of the model's output. The model sometimes exhibits free-form thinking behaviors including: (1) dynamic use of multiple thinking markers (e.g., nested <think></think> blocks), and (2) spontaneous code-switching between languages when conducting reasoning.
>
> ### Novelty of the recipe
> Our pipeline, while retaining the simplicity of vanilla PPO, introduces three scale-critical components that depart markedly from prior PPO implementations—and these deviations are uniquely critical for large-scale reasoning-oriented RL.
>
> ### Hallucination and prompt-length sensitivity
> We benchmark ORZ-32B on MATHIF, which measures instruction-following in mathematical reasoning.
> Correctness is reported without and with explicit constraints (w/o const. vs. w/ const.); the final column gives the relative change when constraints are applied.
> By contrast, all other models in the table lose 4–40% under the same test, showing that ORZ’s alignment maintains and even improves correctness where others falter.
> | model | Correctness w/o const. | Correctness w/ const. | Diff.(%) |
> |-------|------------------------|----------------------|----------|
> | DeepSeek-R1-Distill-Qwen-7B | 65.24 | 48.57 | -25.55 |
> | DeepSeek-R1-Distill-Llama-8B | 59.76 | 36.43 | -39.04 |
> | **Open-Reasoner-Zero-7B** | 52.86 | 51.9 | -1.82 |
> | Qwen3-32B | 72.62 | 70 | -3.61 |
> | DeepSeek-R1-Distill-Qwen-32B | 71.43 | 57.62 | -19.33 |
> | DeepSeek-R1-Distill-Llama-70B | 71.19 | 54.05 | -24.08 |
> | **Open-Reasoner-Zero-32B** | 65.48 | 67.62 | +3.27 |
>
> We hope these additions address the reviewer’s questions about broader behaviour, clarify the source of our efficiency gains, and further demonstrate ORZ-32B’s value to the field.

---

> > ### Comment · Reviewer_wfie · 2025-08-06
> >
> > Thank you for your thorough response. I appreciate the clarifications and the time you took to address my points. I will retain my positive evaluation of the submission.

---

> > > ### Author Response · Authors · 2025-08-06
> > >
> > > Thank you for your continued positive evaluation and for engaging deeply with our work!
> > > We’re pleased our clarifications were useful and will integrate your feedback to further strengthen the final manuscript!

---

### Official Review · Reviewer_5KN5 · 2025-07-03

**Clarity:** 3
**Significance:** 4
**Originality:** 2
**Rating:** 5
**Confidence:** 3

**Summary:**

The authors provide an open-source reproduction of large-scale RL reasoning results, matching the performance of DeepSeek-R1-Zero over four benchmarks. They also provide analysis over design decisions and simplifications the method which reduce the number of training steps.

**Questions:**

Code release would be important.

**Ethical Concerns:**

["NO or VERY MINOR ethics concerns only"]

**Final Justification:**

The paper offers a valuable contribution to the field in terms of open-sourcing. The authors have promised to correct my issues with organization and to actually release the code. I think there is an opportunity for further strengthening the paper with additional analysis and experiments, but this should not be a blocker for acceptance.

**Limitations:**

Yes

**Paper Formatting Concerns:**

No concerns.

**Quality:**

3

**Strengths And Weaknesses:**

Strengths
- Provides open-source experimental methodology to match the performance of a well-known agent. This provides a clear value to the community, in terms of reproducibility and providing a clear framework for further iteration.
- The authors simplify aspects/ablate over the original approach, which meaningfully reduces the number of training steps and may provide valuable insights for practitioners.

Weaknesses
- Organization: In general, while I found the paper clearly written, I do think the paper suffers from poor structure and organization of results. I often had to scroll back and forth to find relevant figures, or explanations of certain results. For example, while Figure 3 is clearly relevant to the design choices in 2.2, it only gets explained 3.2, almost two pages later.
- Although it’s interesting that PPO allows for better advantage estimation on repetitive tokens, the authors do not present any explanation as to why this might be the case. While this observation is clearly useful, understanding this phenomenon might allow for better algorithm design of future work and would strengthen the paper.
- It should not go unmentioned that the novelty is low, given it is mainly centered around a reproduction of prior work. Regardless, I think the open-sourcing and simplifications outweigh this negative.
- No code included with submission.

---

> ### Author Rebuttal · Authors · 2025-07-31
>
> We sincerely thank the reviewer for the positive assessment and valuable feedback. We appreciate the recognition of our work's contribution to the community through open-source methodology and meaningful simplifications. We address each concern below.
> ### Organization and Structure
> Thanks for your valuable feedback on the structure and organization of our results. We fully agree that improving the alignment between figures, their explanations, and relevant sections is crucial for readability. We will reorganize the manuscript in the camera-ready version to ensure better flow between related content. Specifically, we will move the explanation of Figure 3 closer to Section 2.2 where the design choices are introduced, eliminating the need for readers to scroll between sections. We will also improve cross-referencing throughout the paper to enhance readability.
>
> ### Understanding PPO's Superior Advantage Estimation on Repetitive Tokens
> We appreciate the reviewer's interest in understanding why PPO achieve better repetitive token advantage estimation. Our analysis suggests the following mechanism:
> During training, we find that repetitive token sequences reliably signal wrong reasoning paths and failed solutions. The value function quickly learns to give such states low scores, so when the model slips into repetition, the critic usually flags the state with negative advantages and pushes the policy out of the lower value states. Critic-free methods like GRPO lack this salient feedback and miss the cue.
>
> ### Novelty
> While the core algorithm is purposefully minimal, the contribution is the first fully open-source, greatly simplified, efficient reproduction of DeepSeek-R1-Zero. Matching its accuracy with 10× fewer steps, dissecting which components are dispensable, and providing a reusable training harness collectively advance the field's understanding and accessibility of reasoning-oriented RL training.
>
> ### Full open-source release.
> We understand the reviewer's concern about code availability. Unfortunately, NeurIPS 2024 submission guidelines prohibit  sharing URLs or de-anonymizing information during the review period.
> To ensure reproducibility,  we commit to releasing our complete codebase, training data, model checkpoints, and detailed reproduction instructions immediately after double-blind review period. Our submission includes a clear reproducibility statement affirming this commitment.
>
> We hope these clarifications address the reviewer's concerns and reinforce the value of our contributions to the community.

---

> > ### Comment · Reviewer_5KN5 · 2025-08-04
> >
> > Thank you for responding to my questions and concerns. I am happy to keep my original recomendation of acceptance.

---

> > > ### Author Response · Authors · 2025-08-05
> > >
> > > Thank you for your positive assessment and for engaging so thoughtfully with our work.
> > > We appreciate your recommendation for acceptance, and we will incorporate your feedback to further strengthen the final version of the paper!

---

### Official Review · Reviewer_am8n · 2025-07-03

**Clarity:** 3
**Significance:** 3
**Originality:** 3
**Rating:** 5
**Confidence:** 4

**Summary:**

The paper builds on insights from DeepSeek-R1-Zero and introduces an open source and scalabe approach for large scale reinforcement learning from base models. The approach proposed in the work is minimal and demonstrates that vanilla PPO with GAE and simple rule based rewards is sufficient for good performance. The proposed recipe also avoids KL regularization, which simplifies it further. The strengths of the work include open source code, hyper-parameters and training data, instead from model weights alone, and the simplicity of the approach. The main weakness is limited demonstration on math and reasoning benchmarks, and overlap with previous works which have attempted to reproduce DeepSeek-R1-Zero.

**Questions:**

Have you experimented on non Qwen models to explore generalizability?
What modifications would be needed to generalize to code and other domains?
Are there any emergent patterns that are interesting to highlight which haven't been shown by previous works?
Did you use multilingual data in your training set?

**Ethical Concerns:**

["NO or VERY MINOR ethics concerns only"]

**Final Justification:**

The paper is well suited for the conference, and the results presented here would be of interest to the wider community. I recommend acceptance.

**Limitations:**

yes

**Quality:**

3

**Strengths And Weaknesses:**

Strengths:
1. The technique is simple and scalable, doing away with complexities like KL penalty and complex reward function design.
2. The authors have open sourced code, weights and also training data, which fosters reproducibility.
3. The results demonstrate that their recipe is scalable and also more efficient compared to previous works - achieving strong performance with fewer training steps.

Weaknesses:
1. Narrow focus on reasoning and math tasks.
2. While the empirical results are solid, the authors have not explored from a theoretical standpoint why the minimalist approach works so well, and whether it is really superior to previous approaches.

---

> ### Author Rebuttal · Authors · 2025-07-31
>
> Thanks to the reviewer for the careful reading and constructive feedback. We appreciate the acknowledgement of our open-source philosophy and the paper’s practical value, and we address the raised points below.
>
> ### Extension to domains beyond math and reasoning
> Our recipe transfers to new domains (e.g. code, puzzle, instruction-following dialogues) without any modifications. Note that the original submission has already included puzzle-style tasks such as table-join, table-reformat, zebra, and connections alongside math problems. To further demonstrate generalizability, we conducted additional experiments on Qwen-2.5-7B base model with a mixed-domain training distribution (math:code:puzzle:instruction-following = 40%:40%:15%:5%) using the identical ORZ recipe. Training progressed smoothly with consistent increases in both reward and response length. The table below compares benchmark performance between this mixed-domain model and our math-focused model across multiple evaluations.
> | init_model | training domain | train_iter | AIME24 | AIME25 | MATH500 | GPQA_DIA | LCB_V5 | LCB_V6 | IFEVAL | LIVEBENCH | MMLU_TEST | MMLU_PRO | NAIVE_AVG |
> |------------|-----------------|------------|--------|--------|---------|----------|--------|--------|--------|-----------|-----------|----------|-----------|
> | Qwen2.5-7B-Instruct | - | - | 10.10% | 6.04% | 73.80% | 35.70% | 12.70% | 17.14% | 55.82% | 28.83% | 72.66% | 50.91% | 36.37% |
> | Qwen2.5-7B-Base | Math+Code+Puzzle+IF | 500 | 16.46% | 14.69% | 79.00% | 35.89% | 20.79% | 23.14% | 66.27% | 40.51% | 73.81% | 54.06% | 42.46% |
> | Qwen2.5-7B-Base | Math | 500 | 16.20% | 13.13% | 79.60% | 36.71% | 1.30% | 1.72% | 40.27% | 33.99% | 74.67% | 56.08% | 35.37% |
> | Qwen2.5-7B-Base | Math | 700 | 17.50% | 15.00% | 81.00% | 37.85% | 2.15% | 2.29% | 37.62% | 35.97% | 75.04% | 57.80% | 36.22% |
>
> The enlarged task mix delivers substantial gains beyond pure mathematics: IFEVAL, LIVEBENCH, and the logic/code-heavy LCB suites improve sharply, yet the minimalist PPO recipe remains untouched, clear evidence of stronger reasoning transfer.
> This confirms that our recipe generalizes seamlessly to diverse domains without any algorithmic changes.
>
> ### Generalization to non-Qwen model families
> We successfully applied our ORZ recipe to Llama-3.1-8B base model with minimal adaptation (in the above mixed-domain training setting, only adjusting the learning rate from 1e-6 to 5e-7). Training metrics show steady increases in both reward and response length, as shown in the table below:
> | train_iters | 0 | 100 | 200 | 300 | 400 | 500 |
> |-------------|---|-----|-----|-----|-----|-----|
> | response_len | 458 | 230 | 307 | 356 | 546 | 496 |
> | reward | 1.10% | 3.49% | 8.05% | 9.20% | 11.91% | 12.31% |
>
> Offline evaluations show consistent performance improvements across benchmarks as table below. The ORZ-Llama3.1-8B model in the table below corresponds to the checkpoint at iteration 500 (iter500) from applying our ORZ recipe to the Llama3.1-8B-Base model. The NAIVE_AVG is benchmark level of arithmetic mean of all listed benchmarks.
> | model | AIME24 | AIME25 | MATH500 | GPQA_DIA | IFEVAL | LIVEBENCH | MMLU_TEST | NAIVE_AVG |
> |-------|--------|--------|---------|----------|--------|-----------|-----------|-----------|
> | Llama-3.1-8B-Instruct | 2.71% | 0.42% | 39.60% | 27.68% | 54.21% | 19.36% | 62.26% | 29.46% |
> | Llama3.1-8B-Base | 0.16% | 0.00% | 0.29% | 14.29% | 15.96% | 3.56% | 27.83% | 8.87% |
> | ORZ-Llama3.1-8B | 0.52% | 0.16% | 21.93% | 26.70% | 66.96% | 27.17% | 61.21% | 29.24% |
>
> ### Emergent patterns.
> We observe two interesting emergent patterns not well documented previously. First, during the final phases of ORZ-32B training, we observe the emergence of a distinctive reflection pattern where the model spontaneously revisits the problem statement from the beginning when encountering difficulties. Unlike local reflection patterns (e.g., correcting a single step), this global revisiting behavior allows the model to escape local minima by fundamentally reconsidering its approach. Second, while our training focuses on mathematical reasoning, ORZ-32B demonstrates unexpected transfer to logical reasoning in unrelated domains. Without explicit training, the model exhibits structured reasoning patterns in tasks such as academic writing, poetry composing, and general domain quesion and answering.
>
> ### Multilingual data
> Our training data includes multilingual content, with approximately 5% Chinese material from CN K12 datasets in Numina.
>
> ### Theoretical perspective
> We acknowledge the reviewer's interest in theoretical foundations. Crafting a rigorous framework that explains why our minimalist recipe scales reinforcement learning so effectively is an important—but substantial—undertaking. A full treatment would require dedicated analysis beyond the present submission, and we view it as a promising direction for future work.
>
> We will include expanded ablation results and full training-curves in the revision.  We hope these additions address the reviewer’s concerns and clarify the generality, empirical strength, and transparency of Open-Reasoner-Zero.

---

> > ### Comment · Reviewer_am8n · 2025-08-05
> >
> > Thank you for responding to my questions and adding additional results that show generalizability beyond Qwen and explore more tasks. I am changing my recommendation to Accept.

---

> > > ### Author Response · Authors · 2025-08-05
> > >
> > > Thank you for revisiting our work and for the encouraging recommendation!
> > > We’re glad the expanded experiments clarified our method's generalizability.
> > > We will refine the manuscript to include these new results.

---

### Note · Authors · 2025-08-14

We sincerely thank the Area Chair and reviewers for your time, constructive feedback, and active engagement. The discussion substantially improved the clarity, scope, and reproducibility of our work.

There is broad agreement on the paper’s practical value: **a simple, scalable PPO-based recipe** (am8n, 5KN5, 2p6y), **a fully open approach** (am8n, 5KN5, 2p6y), and **thorough experiments** (wfie, Bk7x) that lowers the barrier for large-scale reasoning RL.

During the rebuttal, we expanded evidence beyond the initial submission and addressed the main concerns:
- **Generalization**: We demonstrated transfer to **mixed domains** (math/code/puzzle/IF) without changing the algorithm and showed successful application to **Llama-3.1-8B** with only a minor LR adjustment, along with consistent reward and length growth and offline gains.
- **Beyond benchmark accuracy**: We added open-ended dialogue evaluation (e.g., **ArenaHard v2**) indicating fluent, well-controlled answers. We also reported **MATHIF** results showing robustness under constrained prompting.
- **Reward hacking**: Our exact-match reward on verifiable tasks **mitigates the typical RLHF failure modes**. We supplemented this with qualitative traces exhibiting correct formal reasoning, self-verification, and global “revisit-the-problem” reflection patterns.
- **Organization**: We will restructure figures and cross-references to align design choices with their analyses for smoother reading. We also clarified the continuous colormap used in critical visualizations.

We appreciate this thoughtful review process and hope the additional results and clarifications help with the final decision. We look forward to releasing all artifacts and supporting the community’s exploration of open, scalable reasoning RL.

---

### Decision · Program_Chairs · 2025-09-17

**Decision:**

Accept (poster)

**Comment:**

The paper is an insightful reproduction of DeepSeek-R1-Zero. The reviewers and I unanimously agree it is a value contribution to the community providing a clear study of the key design decisions and hyper-parameter choices behind attaining these results with a variation on PPO.